# Global Transcriptomic Response of *Staphylococcus aureus* to Virulent Bacteriophage Infection

**DOI:** 10.3390/v14030567

**Published:** 2022-03-09

**Authors:** Nikita Kuptsov, Maria Kornienko, Dmitry Bespiatykh, Roman Gorodnichev, Ksenia Klimina, Vladimir Veselovsky, Egor Shitikov

**Affiliations:** Federal Research and Clinical Center of Physical-Chemical Medicine of Federal Medical Biological Agency, 119435 Moscow, Russia; d.bespiatykh@gmail.com (D.B.); grad511@yandex.ru (R.G.); ppp843@yandex.ru (K.K.); djdf26@gmail.com (V.V.); eshitikov@mail.ru (E.S.)

**Keywords:** transcriptome analysis, host–phage interaction, *Staphylococcus aureus*, *Kayvirus*, bacteriophage, RNA-Seq, phage therapy

## Abstract

In light of the ever-increasing number of multidrug-resistant bacteria worldwide, bacteriophages are becoming a valid alternative to antibiotics; therefore, their interactions with host bacteria must be thoroughly investigated. Here, we report genome-wide transcriptional changes in a clinical *Staphylococcus aureus* SA515 strain for three time points after infection with the vB_SauM-515A1 kayvirus. Using an RNA sequencing approach, we identify 263 genes that were differentially expressed (DEGs) between phage-infected and uninfected host samples. Most of the DEGs were identified at an early stage of phage infection and were mainly involved in nucleotide and amino acid metabolism, as well as in cell death prevention. At the subsequent infection stages, the vast majority of DEGs were upregulated. Interestingly, 39 upregulated DEGs were common between the 15th and 30th minutes post-infection, and a substantial number of them belonged to the prophages. Furthermore, some virulence factors were overexpressed at the late infection stage, which necessitates more stringent host strain selection requirements for further use of bacteriophages for therapeutic purposes. Thus, this work allows us to better understand the influence of kayviruses on the metabolic systems of *S. aureus* and contributes to a better comprehension of phage therapy.

## 1. Introduction

*Staphylococcus aureus* is an opportunistic bacterial pathogen that colonizes up to 30% of the human population [1]. Due to variations in virulence factors, *S. aureus* isolates may cause a wide range of diseases, from skin and soft tissue infections to such dangerous diseases as pneumonia, meningitis, and osteomyelitis [2]. Furthermore, the spread of methicillin-resistant Staphylococcus aureus (MRSA) strains complicates disease prognosis by making β -lactam antibiotics ineffective [2]. According to the WHO, the prevalence of MRSA exceeds 20% in some countries [3], and the mortality rate of infections caused by MRSA strains is higher than those caused by methicillin-sensitive *S. aureus* [4]. In addition, multidrug-resistant (MDR) *S. aureus* strains have been increasingly identified in recent years. In European countries, the proportion of *S. aureus* strains resistant to more than one antibiotic accounts for 10% [5].

A promising alternative for the treatment of infections caused by antibiotic-resistant bacteria is phage therapy. Bacteriophages are the most abundant biological objects in the water column of the world’s oceans and are the second-largest biomass component after prokaryotes. Being natural predators, they control the bacterial population. A key feature of bacteriophages is their strict selectivity for bacteria. Phages affect only bacterial host cells as targets and cannot infect eukaryotic cells. Their efficacy has been demonstrated in a variety of animal models [6,7]. In critical situations, due to the lack of effective antibiotics, bacteriophages can be deemed the treatment of last resort [8].

Nowadays, staphylococcal kayviruses (genus *Kayvirus*; subfamily *Twortvirinae*; family *Herelleviridae*) are among the most widely used bacteriophages as therapeutic agents [9]. Bacteriophages of this group are closely related and are obligatorily lytic. Staphylophages have a high lytic capacity, causing lysis of more than 85% of clinical isolates, including MDR isolates [10,11,12,13], and show promising results in the treatment of diseases caused by *S. aureus* [14,15,16].

Although phage therapy is a promising alternative, its regulation and successful mass application require a thorough understanding of the fundamental mechanisms of the interaction between the phage and the host. Advances in omics technologies provide new opportunities in this area, with transcriptome analysis as the most appropriate method to investigate these interactions [17,18,19,20,21,22]. Such studies have previously been carried out for phages of *Pseudomonas aeruginosa* [17,18], *Acinetobacter baumannii* [20], *Bacillus subtilis* [19], *Escherichia coli* [23], *Yersinia enterocolitica* [22], and other bacterial host species. However, in the case of *S. aureus*, data on the transcriptional analysis of the relationship between bacteriophages and their hosts are still insufficient. The existing study on this issue is limited to temperate bacteriophages [24].

In previous reports, we characterized a virulent vB_SauM-515A1 staphylophage isolated from a commercial therapeutic cocktail [12]. The phage demonstrated a wide host range and successfully lysed 85.3% of clinical *S. aureus* strains. Based on transmission electron microscopy and whole-genome sequencing data, vB_SauM-515A1 was identified as a member of the *Herelleviridae* family [25]. The phage genome comprises 238 putative open reading frames (ORFs) and 4 tRNAs (tRNA-Met, tRNA-Trp, tRNA-Phe, and tRNA-Asp) and has no genes for integrases, toxins, or virulence-associated factors. Transcriptional profiling of the vB_SauM-515A1 phage upon *S. aureus* strain SA515 infection revealed that 35 transcriptional units in the genome are regulated by 58 early and 12 late promoters. Early promoters are represented by the strong σ^70^ promoters and control 26 transcriptional units; the late promoters only regulate the expression of four transcriptional units. The remaining five transcriptional units were controlled by both early and late promoters.

In the present study, we aimed to determine changes in the transcriptome response of the host *S. aureus* SA515 strain during vB_SauM-515A1 infection at three time points. Compared to uninfected controls, we found the most significant changes at the onset of infection. Changes were detected in the genes involved in the metabolism of nucleotides and amino acids, as well as in the genes preventing cell death. At the later stages, a significant number of differentially expressed genes (DEGs) were derived from prophage regions and virulence factors. This work can significantly improve our understanding of how phages modify host metabolic systems and may contribute to improved phage therapy.

## 2. Materials and Methods

### 2.1. Strains and Growth Conditions

Isolation and characterization of *Staphylococcus aureus* strain SA515 and lytic phage vB_SauM-515A1 have been described previously [12,25]. In brief, methicillin-susceptible *S. aureus* SA515 was a host for bacteriophage and belonged to ST8 and spa-type t008, according to genotyping schemes [12]. The lytic bacteriophage vB_SauM-515A1 was isolated from a commercial bacteriophage cocktail and is a member of the *Herelleviridae* family [12,25]. The host strain was grown in Luria Bertani (LB) broth or on LB agar plates at 37 °C.

The one-step growth curve of bacteriophage vB_SauM-515A1 on the host strain was performed as previously reported [25]. In brief, SA515 cells at an early exponential phase (OD600 = 0.12) were infected with vB_SauM-515A1 bacteriophage at a multiplicity of infection (MOI) of 0.01. After incubation at 37 °C for 7 min to allow adsorption, the mixture was centrifuged for 3 min at 10,000× *g*. The infected bacterial pellet was resuspended in 10 mL of the LB broth. The aliquots (10 μL) were sampled periodically at 0, 5, 15, 20, 25, 30, 40, 50, 60, and 70 min, where the 0 min point corresponded to the initial infection of the culture with the bacteriophage from the start of infection. Samples were treated with 2% chloroform. The number of vB_SauM-515A1 particles was determined using the double-layer agar plating method.

For the transcriptome analysis, vB_SauM-515A1 bacteriophage was added to SA515 bacterial culture (OD600 = 0.12) at an MOI of 10. Samples were taken at 5, 15, and 30 min after bacteriophage addition. The cells were harvested by centrifugation and immediately frozen at −70 °C. Moreover, the aliquots were taken at 0, 5, 15, 30, 40, 60, and 70 min post-infection to measure the growth curve of host strain by optical density detection and to determine the number of colony-forming units (CFUs). All the aforementioned experiments were carried out in triplicate.

### 2.2. Total RNA Extraction and Sequencing

A detailed technique for RNA preparation and sequencing was described in a previous paper [25]. Briefly, the bacterial cells were disrupted by bead beating in TRIzol reagent (Invitrogen, Carlsbad, CA, USA) and extracted with chloroform. TURBO DNA-free kit (Thermo Fisher Scientific, Waltham, MA, USA) and RNase-Free DNase Set (Qiagen, Hilden, Germany) were used for DNase treatment. RNA cleanup was performed with the RNeasy Mini Kit (Qiagen, Hilden, Germany). Libraries for RNA sequencing were prepared using the NEBNext Ultra II Directional RNA Library Prep Kit (NEB, Ipswich, MA, USA) with prior removal of ribosomal RNA by the RiboMinus Transcriptome Isolation Kit for bacteria (Thermo Fisher Scientific, Waltham, MA, USA). Libraries were sequenced by a high throughput run on the Illumina HiSeq using 2 × 100 bp paired-end reads. The RNA-Seq dataset was deposited to the NCBI under accession number PRJNA659920.

### 2.3. Whole-Genome Sequencing

The genomic DNA of *S. aureus* SA515 was extracted using the QIAamp DNA Kit (Qiagen, Hilden, Germany) following the manufacturer’s instructions. The concentration and quality of the extracted DNA were checked using the Quant-iT DNA Assay Kit, High Sensitivity (Thermo Fisher Scientific Inc., Waltham, MA, USA) and the Agilent DNA High Sensitive Kit (Agilent Technologies, Santa Clara, CA, USA), respectively. Libraries were prepared according to the manufacturer’s instructions using the NEBNext^®^ Ultra™ II DNA Library Prep Kit for Illumina^®^ (NEB, Ipswich, MA, USA). Libraries were subsequently quantified by the Quant-iT DNA Assay Kit, High Sensitivity (Thermo Fisher Scientific Inc., Waltham, MA, USA). Libraries were sequenced by a high throughput run on the Illumina HiSeq using 2 × 100 bp paired-end reads.

### 2.4. Bioinformatics Analysis

The genome assembly of *S. aureus* strain SA515 was performed using Unicycler (v0.4.8) with the default settings (GeneBank accession no. JAKRSL000000000) [26]. The annotation of the assembly was performed with the prokka (v1.14.6) [27]. The sequenced reads were mapped to the *S. aureus* SA515 genome using HISAT2 (v2.2.1) [28]. SAMtools (v1.11) [29] software was used to compress mapped SAM files into BAM and for their subsequent sorting and indexing. Mapping quality and coverage along genes were assessed with QualiMap (v2.2.2) [30], and individual reports were merged with MultiQC (v1.9) [31]. Mapped reads were assigned to genes with featureCounts (v2.0.1) [32]. Differential gene expression analysis was performed using the edgeR (v3.36.0) [33] package for R. Genes, with a false discovery rate (FDR) cutoff of 0.001 and with a fold change log_2_(FC) threshold of |1| (i.e., ≥|2|-fold change) considered to be differentially expressed. Further functional classification of DEGs into Gene Ontology (GO) categories was done using the Panther database (http://www.pantherdb.org, accessed on 11 January 2022); categories were considered enriched with *p*_adj._ ≤ 0.05. Plots were generated within R using ggplot2 (v3.3.2) [34], ggpubr (v0.4.0) [35], ggalt (v0.4.0) [36], ggvenn (v0.1.9) [37], lemon (v0.4.5) [38], cowplot (v1.1.0) [39], and pBrackets (v1.0.1) [40] packages. The prediction of the prophage sequences in the SA515 genome was performed using PHASTER [41].

## 3. Results

### 3.1. Experimental Design of the Study

To assess the bacterial transcriptional response to bacteriophage infection, the virulent bacteriophage vB_SauM-515A1 and the *S. aureus* strain SA515 were chosen. The bacterium and the bacteriophage characteristics as well as the choice of time points for RNA-Seq are described in detail in our previous study [25].

In brief, whole-transcriptome sequencing was performed at three time points (5 min, 15 min, and 30 min), representing different stages of the vB_SauM-515A1 phage lifecycle (Figure 1A). The first two time points corresponded to the host takeover and biosynthesis phases of the phage, while the selection of the last time point was associated with a shift from early gene expression to late genes, the completion of phage assembly, and its release from the host. Meanwhile, analysis of bacterial growth rate at these time points did not reveal that phage affects *S. aureus* culture until the late stage of its life cycle; a reduction in the number of *S. aureus* cells was observed only after 30 min of infection compared to non-infected controls (Figure 1B). However, the plating of infected cells showed a significant decrease in the number of CFUs throughout the entire infection period (Figure 1C).

Prior to the transcriptomic data analysis, whole-genome sequencing of the *S. aureus* SA515 was carried out. The draft genome assembly of *S. aureus* SA515 resulted in 51 contigs with a total length of 2,850,828 bp and an N50 value of 138,814 bp; the longest and the shortest contigs were 491,413 bp and 1210 bp, respectively. A total of 2658 CDSs, three types of rRNA, 53 tRNAs, one tmRNA, and two repeat regions were annotated (Appendix A). Among the predicted CDSs, 2103 genes were assigned a putative function and 555 were annotated as hypothetical proteins.

### 3.2. Influence of Phage Infection on Host Gene Expression

An average of 10 million and 9.3 million high-quality reads were generated for phage-infected and uninfected bacterial cultures, respectively. Subsequently, ~6.9 million (phage-infected cultures) and ~9.2 million (uninfected cultures) reads, per library, were successfully mapped to the *S. aureus* SA515 genome and ~3.8 million reads—to the vB_SauM-515A1 bacteriophage genome. During the infection process, the proportion of reads mapping to the bacterial genome decreased from 87% at 5 min to 42% at 30 min. In turn, the proportion of reads mapping to the phage genome increased from 10% at 5 min to 56% at 30 min (Appendix A). Multidimensional scaling (MDS) of normalized RNA-seq data (three biological replicates and six different conditions) showed a clear separation of samples by condition (Control-Infected) and clustering by time post-infection (Appendix A).

A total of 263 differentially expressed genes (DEGs) were identified (FC ≥ |2|; FDR < 0.001) in the phage-infected host relative to the uninfected host (Appendix A). Among them, 84 and 174 genes were solely down- and upregulated, respectively, while 5 genes changed the direction of the expression pattern. Most of the genes were differentially expressed (*n* = 176; 93↑; 83↓) at the early stage of infection (5 min). At the late stages of infection, the vast majority of DEGs were upregulated (at 15 min [47↑; 0↓] and 30 min [82↑; 8↓]). It is worth noting that DEGs identified at 5 min post-infection mostly did not overlap with DEGs at other time points, whereas at 15 and 30 min post-infection, 39 DEGs were shared (Appendix A).

### 3.3. Shutoff of Host Macromolecular Synthesis in the Early Stages of Infection

To gain further insight into the bacterial response to infection, DEGs at 5 min were used for functional enrichment analysis. GO enrichment analysis classified these genes into 53 enriched GO terms (*p*_adj._ ≤ 0.05), among which, 49 terms corresponded to biological processes (BP), one to molecular functions (MF), and three to cellular components (CC) (Figure 2).

Most of the genes in the BP category were downregulated (78.5%) and were mainly attributed to amino acid and nucleoside metabolic processes. Of them, genes related to nucleoside metabolic processes were assigned to purine and pyrimidine metabolism; two genes (*pyrB* and *pyrC*) of the *pyrABCDEFR* operon and six genes (*purE, purK, purC, purS, purQ*, and *purL*) of the *purEKCSQLFMNHD* operon were downregulated upon bacteriophage infection. In addition, an alternative substrate biosynthesis pathway for purine metabolism was also affected at the level of histidine biosynthesis: products of the downregulated genes *hisH* and *hisF* are responsible for the formation of the 5-aminoimidazole-4-carboxamide ribonucleotide, which is required for further purine synthesis.

Among the genes involved in amino acid metabolism, the most significant changes in expression levels were observed for the genes involved in amino acid catabolism of histidine, proline, alanine, threonine, serine, and arginine. In the case of histidine, in addition to the biosynthesis genes described above, the expression of the genes involved in its degradation with the formation of glutamate (*hutG* and *hutU*) was also reduced. In the case of arginine metabolism, changes in the expression levels of genes involved in the catabolism (*arcA, arcB, arcD*) of this amino acid and those determining its biosynthesis (*argF*) were discovered. Proline catabolism disruption was associated with a decrease in the *fadM* expression level, the product of which converts proline to delta-1-pyrroline-5-carboxylate. This is the primary reaction of the L-glutamate synthesis from L-proline. The change in alanine catabolism during phage infection is associated with the downregulation of the alanine dehydrogenase gene (*ald1*), which is responsible for the synthesis of ammonia and pyruvate from L-alanine. Serine and threonine catabolism was affected by reduced expression of *alsS* (acetolactate synthase), *tdcB* (L-threonine dehydratase catabolic *TdcB*), and *sdaAA* (L-serine dehydratase, alpha chain). At the same time, it should be noted that a reduced expression of these genes may decrease the biosynthesis level of branched-chain amino acids.

### 3.4. Prophage Activation in the Late Stages of Infection

GO enrichment analysis of down- and upregulated DEGs at 15 and 30 min revealed no statistically significant GO terms. However, the significant part of DEGs comprises the genes that were annotated as prophage genes (*n* = 20 for 15 min; *n* = 28 for 30 min).

Three regions corresponding to intact prophages were found in the SA515 genome assembly by PHASTER (Table 1). These phage regions showed genetic similarity with *Staphylococcus phage* phiJB, *S. phage* Sa3, and *S. phage* phi2958PVL, belonging to the temperate staphylophages of the *Siphoviridae* family. It is worth noting that during the infection, only two and five genes for phage region No. 1 and No. 2, respectively, were overexpressed at 30 min post-infection, while for phage region No. 3, a significant number of the genes (*n* = 21) were upregulated at the same time point. The prophage from region No. 3 was identified as a Sa3int phage, the typical sak-carrying phage.

### 3.5. Effect of Bacteriophage Infection on Host Virulence

Considering the prospective use of bacteriophages as antibacterial agents, the effect of bacteriophage vB_SauM-515A1 on the expression of host virulence factors was evaluated. Different virulence factor genes (*n* = 72) were found in the genome of the SA515 strain, including numerous genes of adherence, serine and cysteine proteases, staphylokinase, capsule, and type VII secretion system (Appendix A). Among the found virulence factors, 17 toxin genes were annotated (hemolysin alpha, delta, gamma, enterotoxin A, exotoxins [*set21, set30-set34, set36-set40*]).

Transcriptional data analysis revealed eleven virulence factor-related DEGs (Table 2). At the early stage of infection, three downregulated virulence factor genes were observed: staphylocoagulase (*coa*), triacylglycerol lipase (*lip*), and one exotoxin (*set14*). At the late phage-infection stage, the upregulated virulence genes included three toxin genes (hemolysin gamma and exotoxin), Type VII secretion system genes (*essA, esxA*), and immune evasion factors (*scn, sbi*).

## 4. Discussion

The current work, along with our previous study, describes changes in the gene expression profiles during *S. aureus* SA515 infection with the vB_SauM-515A1 bacteriophage. Previously, we established the transcriptional landscape of the vB_SauM-515A1 bacteriophage and demonstrated that most of the genes are constitutively expressed throughout the infection. Only by the 30th minute of infection was late gene transcription observed, without evident shutdown of the early genes [25]. Here we showed that only 10% of *S. aureus* genes altered their expression, which is consistent with previously published reports on various phage-infected bacteria, e.g., 7.1% of host genes were differentially expressed in PaP1-infected (*Myoviridae*) *Pseudomonas aeruginosa* [18], 2.7% in φ29-infected (*Podoviridae*) *Bacillus subtilis* [19], and 15.6% in φAbp1-infected (*Autographviridae*) *Acinetobacter baumanii* [20]. Notably, the strongest response of *S. aureus* was registered at the onset of infection (5 min post-infection), when a relatively equal number of genes were up- and downregulated. Intriguingly, most of the phage genes were also expressed at 5 min post-infection; this may suggest a phage-controlled regulation of bacterial gene expression.

During the onset of infection, the nucleic acid metabolism underwent the most significant changes, which is typical for various phage-infected bacterial species [42]. The purine biosynthesis pathway is crucial for cell growth and is involved in bacterial survival and virulence [43]. Moreover, purine synthesis is associated with (p)ppGpp alarmone level [44]. The accumulation of (p)ppGpp can lead to the inhibition of transcription, translation, and premature cell death [45,46], following which the bacteriophage does not multiply effectively within the cell. In turn, the studied vB_SauM-515A1 bacteriophage is able to reduce the level of (p)ppGpp alarmone by expressing the *hmzG* gene. The *hmzG* gene is a homolog of the *mazG* gene of the MazFG toxin-antitoxin system, which mediates programmed cell death. MazF protein degrades mRNA, whereas MazG protein inhibits MazF and (p)ppGpp synthesis [47,48]. In addition to the described mechanism, the found changes in the expression of LytSR regulatory system genes ([*lytR*↑ at 5 min], IrgAB [*lrgB_1*↑ at 30 min], and SarV [*sarV*↓ at 5 min]) might also be involved in preventing cell death. The two-component LytSR system is a negative regulator of autolysis and biofilm formation in *S. aureus* [49] and directly affects *lrgAB* operon transcription, which, in turn, encodes an anti-holin-like protein, preventing autolysis [50]. In a recently published study, analysis of the *sarV* mutant indicated that this gene regulates extracellular and intracellular murein hydrolase activity and constitutes an important “hub” for the control of autolysis by *mgrA* and *sarA*. *S. aureus* strains were highly susceptible to lysis when *sarV* was overexpressed [51].

In addition to the affected nucleic acid metabolism, GO enrichment analysis revealed notable changes in the expression level of amino acid metabolism genes, which is also consistent with previously published data [18,20,21]. Differences in amino acid gene expression profile caused by phage infection are generally associated with the fact that bacteriophages can have different amino acid requirements than uninfected bacterial cells [21]. Furthermore, since *S. aureus* has been shown to use amino acids as a carbon source, degrading them to the metabolic intermediates, namely pyruvate (from alanine, serine, glycine, threonine, and cysteine), 2-oxoglutarate (from glutamate, glutamine, histidine, arginine, and proline), and oxaloacetate (from aspartate and asparagine) [52], the depletion of the free amino acid pool can lead to a slowdown in the metabolism and growth of the bacterial cell.

Analysis of subsequent time points revealed a rather different expression profile compared to the onset of infection: the bulk of the DEGs at the 15th and 30th minutes were upregulated and mostly overlapped between the two time points. It is worth noting that a significant part of the upregulated DEGs comprised one of the three found in the genome prophages and virulence factors. The discovered features raise the issue of the use of clinical strains as propagation hosts, as they are likely to contain prophages and can produce toxin genes and determinants of antibiotic resistance, e.g., in *S. aureus* prophages encode such toxins as Panton–Valentine leukocidin, exfoliative toxins Eta and Etb, etc. [53]. In connection with the induction of the prophage, the probability of horizontal toxin gene transfer in the population of the pathogenic bacterium increases; therefore, a more detailed study of the possibility of prophage induction when using virulent bacteriophages for therapeutic purposes is necessary. At the same time, overexpression of certain virulence factors genes, which are not associated with prophage regions, was discovered for the first time for virulent bacteriophage–bacteria pairs; this finding imposes additional requirements for phage purification methods.

## 5. Conclusions

In conclusion, the present study represents the general description of the clinical *S. aureus* strain transcriptome-wide response to virulence phage infection. The main changes in the host transcriptome are related to the metabolism of key cellular macromolecules as well as the regulation of cell death. In addition, the effect of a virulent bacteriophage on the expression of prophage and virulence factor genes was also shown. We believe that our findings can provide a better understanding of the fundamental principles of interaction between virulent staphylophages and their hosts and may also lay the foundation for the safe and rational use of virulent bacteriophages for therapeutic purposes.

## Figures and Tables

**Figure 1 viruses-14-00567-f001:**
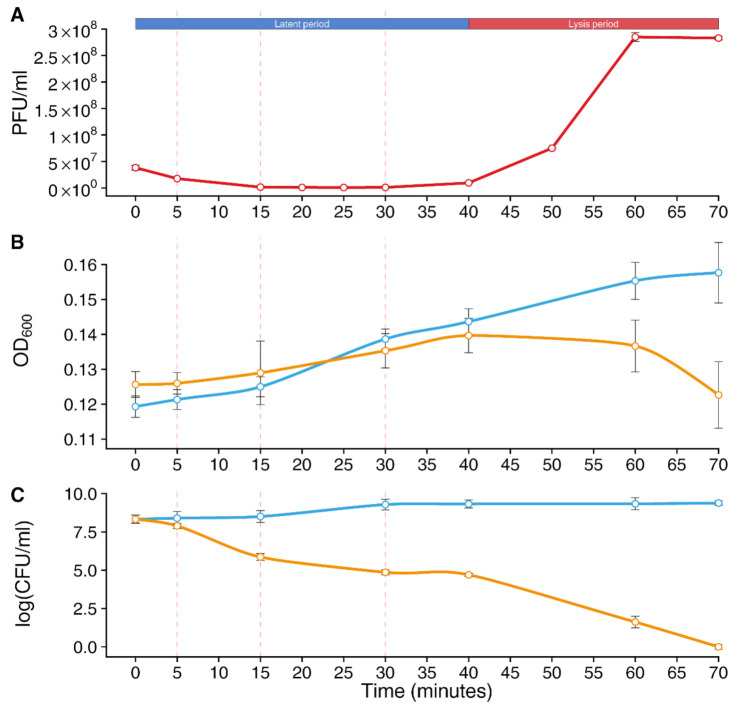
Analysis of the replication cycle of a bacteriophage and its effect on the bacterial growth. (**A**) The one-step growth curve of vB_SauM-515A1 bacteriophage, modified from [25]. (**B**,**C**) Growth inhibition curves of SA515 strain by the vB_SauM-515A1 phage. Phage-mediated lysis of bacteria was monitored by measuring the OD_600_ values and colony-forming units/mL (CFU/mL) throughout the infection. The blue color corresponds to uninfected cells of SA515 strain, and yellow to phage-infected cells. Dotted lines indicate the three sampling time points selected for the transcriptomic analysis.

**Figure 2 viruses-14-00567-f002:**
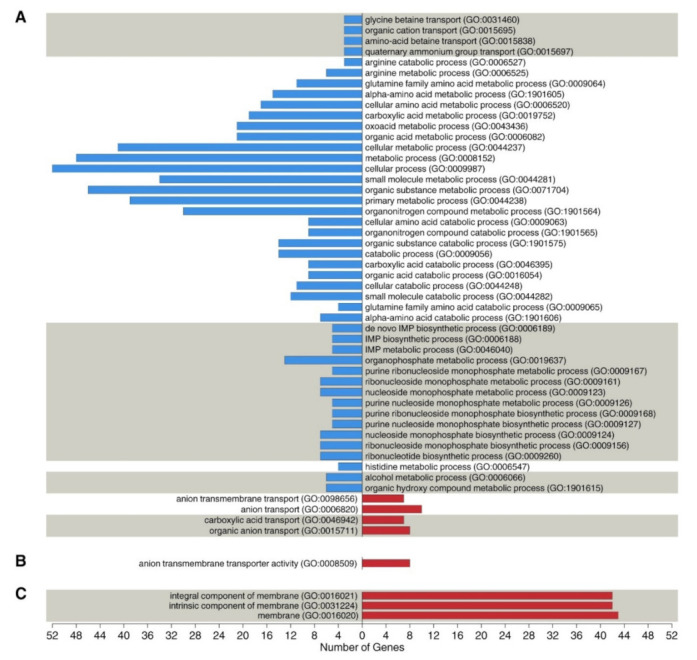
Gene Ontology (GO) enrichment analysis of host differentially expressed genes (DEGs) (up- and downregulated genes) in the early stage of infection (5 min). (**A**) biological processes; (**B**) molecular function; (**C**) cellular component. Enriched GO terms for DEGs colored by red and blue. Red bars indicate terms enriched in the upregulated DEGs. Blue bars represent terms enriched in the downregulated DEGs.

**Table 1 viruses-14-00567-t001:** Prediction of prophage regions in the *S. aureus* strain SA515 genome using PHASTER.

Region	Contigs	Possible Phage Match and Its Characteristics		
		**Phage**	**Phage Size, bp**	**Phage CDSs**
1	Saur515_contig_7 Saur515_contig_22	*Staphylococcus phage* phiJB (NC_028669)	43,012	70
2	Saur515_contig_5	*Staphylococcus phage* phi2958PVL (NC_011344.1)	47,342	59
3	Saur515_contig_21 Saur515_contig_28	*Staphylococcus phage* Sa3 (OC8) (LC129040.1)	42,984	60

**Table 2 viruses-14-00567-t002:** Expression of SA515 genes related to virulence after phage infection.

Virulence Factor	Gene	Fold Change *		
		5 min	15 min	30 min
**Downregulated genes**				
toxin superantigen-like protein, exotoxin 14	*set14*	**−2.09**	1.10	−1.36
staphylocoagulase	*coa*	**−2.31**	−1.00	1.34
triacylglycerol lipase	*lip*	**−2.20**	−1.19	−1.09
**Upregulated genes**				
gamma-hemolysin component C precursor, HlgB	*hlgB*	−1.53	**2.53**	**4.31**
gamma-hemolysin component C precursor, HlgC	*hlgC*	−1.36	**2.98**	**4.00**
superantigen-like protein, exotoxin set40	*set15*	−1.27	1.28	**2.85**
fibrinogen-binding protein	*efb*	−1.26	1.66	**2.99**
ESAT-6/WXG100 family secreted protein EsxA/YukE	*esxA*	1.14	**3.04**	**4.99**
protein secretion system EssA	*essA*	1.00	1.29	**2.04**
immunoglobulin G-binding protein SBI	*sbi*	−1.14	1.46	**2.62**
staphylococcal complement inhibitor SCIN	*scn*	−1.00	1.68	**3.11**

* Values in bold meet the corresponding selection criteria (FC ≥ |2|; FDR < 0.001).

## Data Availability

Not applicable.

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
