# Peer review of "Global Transcriptomic Response of Staphylococcus aureus to Virulent Bacteriophage Infection"

_viruses, 2022, doi:10.3390/v14030567_

Round 1

Reviewer 1 Report

Kuptsov et al present transcriptional response of S. aureus host infected by virulent myophage. The authors continue with their previous work published in 2020 in Viruses (Kornienko et al), where the authors summarized the transcription of the phage.

The presented study thus completes the information in phage-host transcriptomic interaction.

Question 1: The authors identified three putative prophages in the S. aureus SA515 genome. How did they determine their precise length (Table 1), when the prophage regions are spanning more then one contig? Table 1 is confusing, since it is not clear if columns Phage Size, bp and Phage CDSs describe the detected prophage regions in SA515 genome or the Most Common Phage.

Question 2: Are the prophage inducible and viable? Is it possible to induce the detected phage from the host genome, so they would create phage particles?

Question 3: What were the differences between biological replicates? Please provide some basic statistical visualizations?

Question 4: Figure 1: How would the authors explain the decrease of CFU, which is not corresponding to OD600 measurement?

In Figure 2 the authors present results of the GO enrichment analysis. How many DEGs actually appear in the analysis? Some of the categories probably contain the same genes (e.g. Glycine betaine transport and amino-acid betaine transport; de novo IMP biosynthesis, IMP biosynthetic process, IMP metabolic process). Could the authors clarify Figure 2?

The sequenced and annotated genome of S. aureus SA515 should be publicly available, so the experiment could be repeated.

line 177: The authors found 4 rRNAs. In the Methods section is missing, how did they determine the number of the rRNAs. The genome has 51 contigs and the rRNAs are usually at the ends of the contigs.

line 142: Did the authors use for the differential expression analysis also the tRNA and rRNA genes? 

Iine 97-110: It is unclear what time the authors mean with "after infection." In methods section authors describe adsorption time 7 minutes, subsequent centrifugation and then sampling time 0. Please clarify.

Author Response

Thank you very much for the valuable comments on our manuscript. We have carefully considered all your suggestions. Our answers to these questions are as follows in attachment file. Please see the attachment.

Reviewer 2 Report

Bacteria that are resistant to the most commonly used antibiotics are a big problem nowadays. Therefore, the subject of the manuscript is totally topical and relevant.

The use of bacteriophages can be considered as an important alternative to combat these bacteria, and a better understanding of how it works and which genes are involved in the infectious processes is necessary.
I have found no objections to the publication of this work as found. I would only check the abbreviations of words that are not described the first time they appear, for example, GO on line 146.

Author Response

We sincerely thank you for this positive feedback.

Point 1: I have found no objections to the publication of this work as found. I would only check the abbreviations of words that are not described the first time they appear, for example, GO on line 146.

Response 1:This discrepancy has been fixed.